# Evaluating socioeconomic inequalities in influenza vaccine uptake during the COVID-19 pandemic: A cohort study in Greater Manchester, England

Ruth Elizabeth Watkinson[1]*, Richard Williams[2], Stephanie Gillibrand[1], Luke Munford[1], Matt Sutton[1]

**1** NIHR Applied Research Collaboration Greater Manchester / Health Organisation, Policy, and Economics (HOPE) Group, Centre for Primary Care and Health Services Research, University of Manchester, Manchester, United Kingdom, **2** NIHR Greater Manchester Patient Safety Translational Research Centre and NIHR Applied Research Collaboration Greater Manchester, Manchester Academic Health Science Centre, The University of Manchester, Manchester, United Kingdom

* ruth.watkinson@manchester.ac.uk

**Data Availability Statement:** The patient data used in this study cannot be shared publicly. The legal basis for use of patient data in this study was

## Abstract

### Background

There are known socioeconomic inequalities in annual seasonal influenza (flu) vaccine uptake. The Coronavirus Disease 2019 (COVID-19) pandemic was associated with multiple factors that may have affected flu vaccine uptake, including widespread disruption to healthcare services, changes to flu vaccination eligibility and delivery, and increased public awareness and debate about vaccination due to high-profile COVID-19 vaccination campaigns. However, to the best of our knowledge, no existing studies have investigated the consequences for inequalities in flu vaccine uptake, so we aimed to investigate whether socioeconomic inequalities in flu vaccine uptake have widened since the onset of the COVID-19 pandemic.

### Methods and findings

We used deidentified data from electronic health records for a large city region (Greater Manchester, population 2.8 million), focusing on 3 age groups eligible for National Health Service (NHS) flu vaccination: preschool children (age 2 to 3 years), primary school children (age 4 to 9 years), and older adults (age 65 years plus). The sample population varied between 418,790 (2015/16) and 758,483 (2021/22) across each vaccination season. We estimated age-adjusted neighbourhood-level income deprivation-related inequalities in flu vaccine uptake using Cox proportional hazards models and the slope index of inequality (SII), comparing 7 flu vaccination seasons (2015/16 to 2021/22). Among older adults, the SII (i.e., the gap in uptake between the least and most income-deprived areas) doubled over the 7 seasons from 8.48 (95% CI [7.91,9.04]) percentage points to 16.91 (95% CI [16.46,17.36]) percentage points, with approximately 80% of this increase occurring during the pandemic. Before the pandemic, income-related uptake gaps were wider among

defined in the national Control of Patient Information (COPI) notice, which gives National Health Service (NHS) organisations a legal requirement to share data for the purposes of the COVID-19 response and COVID-19-related research. A strict governance process involving stakeholders groups (data controllers, healthcare professionals, patients and members of the public, and researchers) exists for granting researchers access to Greater Manchester Care Record data. For further details please see https://gmwearebettertogether.com/gm-care-record/ or contact GMCR-ops@manchester.ac.uk. All codes, algorithms, and code set validation used to define the populations, outcomes, exposures, and covariates can be found here: https://github.com/rw251/gm-idcr/tree/master/projects/025%20-%20Watkinson.

**Funding:** This study was supported by National Institute for Health and Care Research (NIHR) (grant NIHR200174; to REW, RW, SG, LM & MS, grant PSTRC-2016-003; to RW). The views expressed are those of the authors and not necessarily those of the NIHR or the Department of Health and Social Care. The funders had no role in study design, data collection and analysis, decision to publish, or preparation of the manuscript.

**Competing interests:** I have read the journal's policy and the authors of this manuscript have the following competing interests: MS is a National Institute for Health and Care Research (NIHR) Senior Investigator, whose collective role is to: contribute significantly to NIHR as a senior leader; lead in training and development of NIHR's people; act as an ambassador for the NIHR in the wider system; demonstrate research excellence; contribute to growth; integrate patient, care users, carers and public involvement and engagement in research; and play a leading role in NIHR's contribution to growth. Each Senior Investigator receives a discretionary award of £20,000 per year of appointment to fund activities that support their research. Within the last five years, MS has had the following roles for NIHR: National Lead for Economics in the NIHR Applied Research Collaborations, Chair of the Economics Workstream for the NIHR Methodology Incubator, Chair of the NIHR Economics Group; Member of the National Institute for Health and Care Excellence (NICE) Implementation Strategy Group; Member of the Rapid Service Evaluation Team 2023 Committee; Member of the School for Social Care Phase IV Committee; Deputy Chair of the NIHR Advanced Fellowships Panel; Deputy Chair of the NIHR Health Services and Delivery Research Funding Committee; Member of the NIHR Clinician

children, ranging from 15.59 (95% CI [14.52,16.67]) percentage points to 20.07 (95% CI [18.94,21.20]) percentage points across age groups and vaccination seasons. Among pre-school children, the uptake gap increased in 2020/21 to 25.25 (95% CI [24.04,26.45]) percentage points, before decreasing to 20.86 (95% CI [19.65,22.05]) percentage points in 2021/22. Among primary school children, inequalities increased in both pandemic years to reach 30.27 (95% CI [29.58,30.95]) percentage points in 2021/22. Although vaccine uptake increased during the pandemic, disproportionately larger increases in uptake in less deprived areas created wider inequalities in all age groups. The main limitation of our approach is the use of a local dataset, which may limit generalisability to other geographical settings.

## Conclusions

The COVID-19 pandemic led to increased inequalities in flu vaccine uptake, likely due to changes in demand for vaccination, new delivery models, and disruptions to healthcare and schooling. It will be important to investigate the causes of these increased inequalities and to examine whether these increased inequalities also occurred in the uptake of other routine vaccinations. These new wider inequalities in flu vaccine uptake may exacerbate inequalities in flu-related morbidity and mortality.

## Author summary

### Why was this study done?

- There are known socioeconomic inequalities in flu vaccine uptake, with uptake lower in more deprived neighbourhoods compared to less deprived areas.

- During the Coronavirus Disease 2019 (COVID-19) pandemic, there were many changes that may have affected flu vaccine uptake, including changes in flu vaccine eligibility and delivery, extensive vaccine messaging and debate linked to the COVID-19 mass vaccination campaign, and widespread disruption to healthcare services.

- However, to the best of our knowledge, no existing studies had analysed whether the COVID-19 pandemic was linked to changes in socioeconomic inequalities in flu vaccine uptake.

### What did the authors do and find?

- We used health records for the population of Greater Manchester, a large metropolitan region in the North West of England, to look at changes in socioeconomic inequalities in flu vaccine uptake before and during the COVID-19 pandemic.

- We focused on young children and older adults, as these age groups are at higher risk of severe outcomes from flu infection and are eligible for annual flu vaccination provided free at the point of service.

Scientist Fellowship Panel; Member of the Department of Health Policy Research Programme Board; and Member of five Study Steering Committees. All other authors have declared that no competing interests exist.

**Abbreviations:** ARC-GM, Applied Research Collaboration GM; CI, confidence interval; COVID-19, Coronavirus Disease 2019; GM, Greater Manchester; GMCR, Greater Manchester Care Record; GP, general practice; HInM, Health Innovation Manchester; HR, hazard ratio; IDACI, income deprivation affecting children index; IDAOPI, income deprivation affecting older people index; IMD, index of multiple deprivation; LSOA, lower layer super output area; NHS, National Health Service; NIHR, National Institute of Health and Care Research; SARS-CoV-2, Severe Acute Respiratory Syndrome Coronavirus 2; SII, slope index of inequality.

- We found that the difference in uptake between the least and most deprived areas increased markedly during the pandemic years (2020/21 and 2021/22), with the greatest increase in inequalities among older adults.

## What do these findings mean?

- The COVID-19 pandemic led to increased socioeconomic inequalities in flu vaccine uptake, likely due to changes in demand for and access to flu vaccination.

- It will be important to better understand the causes of these increased inequalities and to investigate whether inequalities have also increased across other sociodemographic factors and for uptake of other routine vaccinations.

- A limitation of this study is the use of a local dataset for Greater Manchester, so although this is a relatively diverse region, our results may not be generalisable to other areas.

## Introduction

Seasonal influenza (flu) infection is associated with high demand for primary care consultations, tens of thousands of hospital admissions, and excess mortality each winter [1–3]. Flu vaccination provides protection against severe clinical flu presentation [2,3], so vaccine uptake is an important determinant of outcomes from flu infection.

In England, the National Health Service (NHS) provides healthcare free at the point of service, and this includes annual flu vaccination for groups at higher risk of severe outcomes from flu infection (young children and adults aged 65 years plus, groups with certain preexisting health conditions, or pregnant people [3]). Despite this free vaccination offer, previous studies show substantial socioeconomic inequalities in flu vaccine uptake prior to the Coronavirus Disease 2019 (COVID-19) pandemic [4,5]. Vaccine uptake was lower among those living in more deprived neighbourhoods, despite higher rates of hospitalisation due to flu infection in deprived areas [4,5]. Possible explanations for these observed inequalities include differential health knowledge and information, with understanding of the risks of flu infection, perceived vulnerability, and concerns about flu vaccine side effects each potentially differing across socioeconomic groups [5].

Pre-COVID-19 pandemic, flu vaccine uptake among older adults was stable, varying between 71% and 75% for over a decade [6]. However, uptake among adults increased to over 80% in each of the 2 winter seasons following the onset of the COVID-19 pandemic, suggesting a link between the pandemic period and flu vaccine uptake [6]. Flu vaccine uptake among preschool and school aged children has consistently been lower than among older adults, ranging from approximately 30% to 60% prepandemic [7,8].

The COVID-19 pandemic was associated with several factors that may impact overall flu vaccine uptake and uptake inequalities. Firstly, there was widespread but unequal disruption to NHS services and to schools (a key site where vaccinations are delivered to children), which may have affected access to flu vaccination [9,10]. There were also changes to flu vaccine delivery in some areas during 2021/22, with optional coadministration of flu and COVID-19 vaccines for eligible adults [11]. Alongside differences in access, attitudes towards flu vaccination

may have changed. For example, concerns about COVID-19, particularly the risks of coinfection with flu [12], may have increased people's perceived need for flu vaccination. The high-profile COVID-19 mass vaccination programme, accompanied by extensive public health messaging and substantial misinformation and disinformation, may also have affected attitudes [13–15]. Concerns about the risk of side effects from vaccination may also have increased, after rollouts of several COVID-19 vaccines were restricted or suspended in some countries following reported links to blood clots and myocarditis [16–19].

To the best of our knowledge, there are no published studies or grey literature that investigate whether inequalities in flu vaccine uptake have changed since the COVID-19 pandemic. We use population-wide data from a large metropolitan area (Greater Manchester, population 2.8 million) to analyse socioeconomic inequalities in flu vaccine uptake before and during the COVID-19 pandemic. Although Greater Manchester is mostly a major urban conurbation, it also encompasses some smaller urban towns as well as some rural towns and villages [20]. Greater Manchester is also a socioeconomically diverse area, including some of the least deprived neighbourhoods nationally, despite overall higher deprivation levels than the national average [21]. For these reasons, Greater Manchester is a useful setting to study changes in socioeconomic inequalities in vaccine uptake.

## Methods

### Ethics statement

The GMCR Research Governance Group approved this project (Ref: IDCR-RQ-25, approval date: 04 February 2021) based on the control of patient information (COPI) notice, which required NHS organisations to share data for the COVID-19 response. The patient data were deidentified, so informed consent was not required. We used the REporting of studies Conducted using Observational Routinely collected Data (RECORD) guidelines to write up this study (S1 RECORD).

### NHS England institutional context

Annual NHS flu vaccination is available to at-risk adults and preschool children (aged 2 to 3 years) free at the point of service at their general practice (GP) surgery [22]. Alternatively, eligible adults can access flu vaccination at community pharmacies [22]. NHS flu vaccination is also available to children aged between 4 and 11 years, but this is delivered through primary schools [23]. Adults are usually given an injected inactivated flu vaccine, while children generally receive a live attenuated nasal spray vaccine [22].

### Data collection

We extracted data from the Greater Manchester Care Record (GMCR) on 6 October 2022. The GMCR has almost population-level coverage within Greater Manchester (GM), holding data on approximately 3 million patients registered with GP surgeries [24]. The GMCR contains all patients in GM who were alive on 1 February 2020. There is some survival bias, as some localities in GM do not provide data on patients who died prior to this date.

### Populations

We restricted analysis to people resident within GM (3,021,322 individuals) to avoid using potentially out-of-date records from individuals who have left the region and may have re-registered with a non-GM GP surgery. There were no known missing data on flu vaccine uptake, clinical vaccination eligibility, or age. We excluded 2,107,742 people who were not eligible for

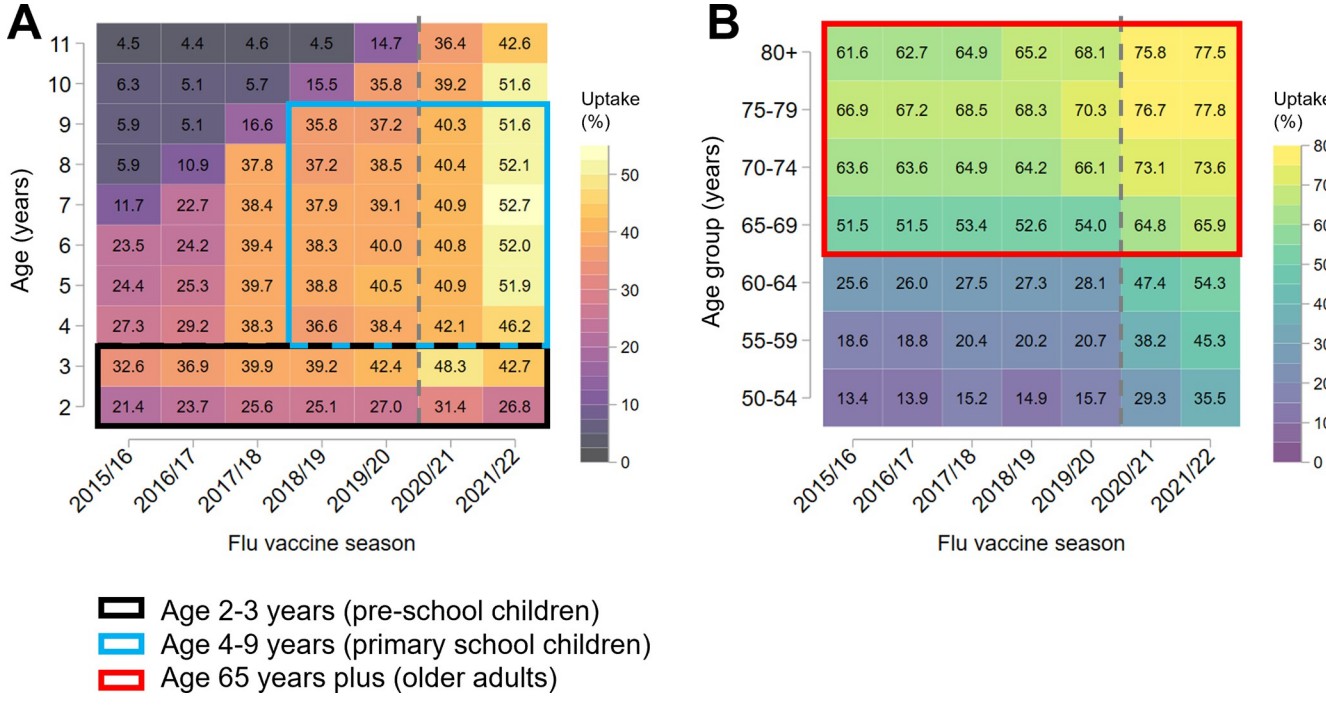

**Fig 1.** Percent flu vaccine uptake by age and flu vaccination season among (**A**) children and (**B**) older adults. The main study populations and seasons are highlighted with coloured boxes as indicated in the legend. The vertical dashed grey line indicates the onset of the pandemic.

NHS flu vaccination on age criteria at any point during the study period, leaving 913,580 individuals for the main analysis. The sample varied between vaccination seasons based on eligibility (age) and deaths (S1 Fig).

Trends in flu vaccine uptake among school-age children are complex, as there has been a piloted and phased rollout [23]. We therefore focused on a subset of ages and seasons as indicated in Fig 1. In the most recent season (2021/22), NHS flu vaccination was also available to older children (aged 11 to 16 years) through schools [23]. However, since uptake data are available for only one season, we excluded this group from the main analysis. We were unable to reliably identify people eligible for NHS flu vaccination due to pregnancy or preexisting health conditions over time, so the main analysis focused on people eligible for vaccination due to age.

## Variables

**Outcomes.** The primary outcome was flu vaccination for the 2015/16 to 2021/22 vaccination seasons (1 September to 31 March the following year). We included date of death (all causes) to identify censoring events. Outcomes appear in GM residents' GMCR record for NHS vaccination across any setting (GP, pharmacy, or school) and whether they occur within GM or elsewhere. Awareness of the COVID-19 pandemic in the United Kingdom (UK) emerged towards the end of the 2019/20 vaccination season, so we consider the 2020/21 and 2021/22 vaccination seasons to have taken place since the start of the pandemic.

**Exposures.** The primary exposure was income deprivation, using the 2019 income deprivation affecting older people index (IDAOPI) rank and income deprivation affecting children index (IDACI) rank from the Ministry of Housing, Communities, and Local Government website at the lower layer super output area (LSOA) level [21]. IDAOPI and IDACI are age-specific income domain components of the index of multiple deprivation (IMD) [21].

Specifically, these measures are derived from the proportion of children aged 0 to 15 years (IDACI) or older adults aged 60 years plus (IDAOPI) in each LSOA who are living in income-deprived households, which is defined by receipt of income-assessed benefits or tax credits with an equivalised income of less than 60% of the national median (before housing costs) [25]. These statistics are based on administrative records held by the Department for Work and Pensions and His Majesty's Revenue and Customs [25].

**Covariates.** The main covariate was age, recorded as age in years at the index date (1 February 2020). Children are eligible for vaccination based on their age before the start of each vaccination season (1 September), whereas adults are eligible based on their age at the end of the season (31 March), much closer to the index date. Age was adjusted for each vaccination season, using the lower bound of possible age during the vaccination season for children and the upper bound for adults to minimise misspecification of vaccine eligibility. Practically, this means that children aged "2" may have been aged 2 or 3 years, during the vaccination season, while adults aged "65" may have been aged 64 or 65 years. In regression analysis, age in years was used for children, and age grouped into 5-year age bands up to 80 years plus was used for adults. We included clinical eligibility only in sensitivity analysis, as it was only available for 2021/22 [26].

## Statistical analysis

**Analysis plan and changes.** The hypothesis for this study was that changes associated with the onset of the COVID-19 pandemic may have widened socioeconomic inequalities in flu vaccine uptake. There was no formal prospective protocol, but in advance of any analysis, we planned to test this by using area-level IMD decile as the socioeconomic measure and investigating both relative inequalities using age-adjusted Cox proportional hazards models for each flu vaccination season, and absolute inequalities using age-standardised uptake. We later added the slope index of inequality (SII) analysis following feedback from colleagues. At the suggestion of one of the reviewers, we added supplementary analysis stratified by sex and added results unadjusted by age at the request of the editor. We initially planned to focus only on older adults, as we were not aware data for people aged under 16 years was available in this dataset. However, after discussion with data engineers, we also included children who were eligible for flu vaccination. There were no data-driven changes in our approach to the analysis.

**Main analysis.** We estimated associations between income deprivation decile and time-to-vaccination (from 1 September each season) using the least deprived decile as the reference group and adjusting by age, using separate Cox proportional hazards models for each season. Death was included as a censoring event. We used the Breslow method to handle ties, with standard errors corrected for heteroscedasticity. We reported results as hazard ratios (HRs) with 95% confidence intervals (CIs). For clarity, we did not report p-values as these refer to within-season comparisons while most comparisons made in the text are between vaccination seasons. We confirmed that the proportional hazards assumption was reasonable by visual inspection of log(−log[survival]) versus log(time) plots (S2 Fig). We used a time-to-event approach, rather than the binary outcome of vaccination received, because delayed vaccination is also a component of the overall vaccine uptake inequality, with earlier vaccination able to offer protection throughout the peak of the winter flu season.

We estimated age-standardised vaccine uptake for each deprivation decile at the end of each vaccination season, excluding those who died before or during each season and using direct standardisation to the total sample population. We estimated absolute inequalities using SII [27] adjusted by age group, using the fractional rank of each LSOA (as opposed to deciles) with LSOAs ranked in descending order of deprivation.

**Sensitivity analysis.** We first repeated the main analysis (Cox regression) without adjustment by age. We then restricted the main analysis sample to those who remained alive throughout all 7 vaccination seasons. We also reestimated results using overall IMD as an alternative measure of deprivation [21]. We then reestimated results excluding people at the limit of age eligibility in each group (i.e., aged 3, 9, or 65 years) to assess potential bias due to misspecification of eligibility due to using age in years (rather than date of birth). We also reestimated 2021/22 results adding in individuals who were clinically eligible for flu vaccination, and those aged 9 to 16 years and 50 to 64 years to match the expanded age-based eligibility during 2021/22 [26]. In addition, we reestimated the main results stratifying by sex (male or female).

All statistical analysis used Stata 16.1.

## Public and community involvement and engagement

This study was part of a larger flu and COVID-19 Vaccine Equity Project. We held several public discussion groups with diverse members of the GM community in partnership with the National Institute of Health and Care Research (NIHR) Applied Research Collaboration GM (ARC-GM) panel and Health Innovation Manchester (HInM) forum as described previously and worked collaboratively with a Public Advisory Group throughout the project [24].

## Results

Among those eligible for flu vaccination based on age during the 2021/22 vaccination season, there were 70,419 children aged 2 to 3 years (referred to as preschool children); 233,277 children aged 4 to 9 years (referred to as primary school children); and 454,787 adults aged 65 years plus (referred to as older adults) (Table 1). Income deprivation was higher in GM than the England average, with approximately 24% of preschool (16,612) and primary school (55,478) children living in the 10% most deprived neighbourhoods nationally. Older adults in GM also disproportionately lived in income-deprived areas (66,568, 14.6%) (Table 1). Flu vaccine uptake was higher in older age groups, with 72.8% of older adults (331,092), 51.1% of primary school children (119,223), and 34.9% of preschool children (24,575) receiving flu vaccination.

Fig 1 shows flu vaccine uptake by age group and vaccination season. The staggered rollout of flu vaccination across children aged 4 to 11 years is visible in the pattern of gradually increasing uptake (Fig 1A). Similarly, uptake among those aged 50 to 64 years increased markedly after eligibility was expanded to this age group in the 2020/21 season (Fig 1B). To compare trends in inequalities over time, we applied vaccine eligibility criteria that were consistent over the study period as indicated by the highlighted boxes in Fig 1. We included preschool children (aged 2 to 3 years) and older adults aged 65 years plus. Given the staggered vaccination rollout in primary schools, we used a subset of primary school ages (aged 4 to 9 years) over a shorter range of seasons (2018/19 to 2021/22).

### Relative inequalities

There were income-related inequalities in flu vaccine uptake across all age groups and seasons, with higher income deprivation associated with lower vaccine uptake (Fig 2, S1, S3, and S5 Tables). Across prepandemic vaccination seasons, uptake inequalities between the most and least income-deprived deciles were greatest among preschool children (HR range 0.52 (95% CI [0.50,0.54]) to 0.54 (95% CI [0.52,0.57])), followed by primary school children (HR range 0.59 (95% CI [0.58,0.61]) to 0.61 (95% CI [0.60,0.63])), with more moderate inequalities among older adults (HR range 0.77 (95% CI [0.76,0.79]) to 0.82 (95% CI

**Table 1. Baseline study population statistics.** Population for 2021/22 season is shown as an illustrative example.

| | Age 2 to 3 years (Preschool children) | Age 4 to 9 years (Primary school children) | Age 65 years plus (Older adults) |
|---|---|---|---|
| **Total population** $N$ | 70,419 | 233,277 | 454,787 |
| **Age** (years) *median (IQR[1])* | 3 (2,3) | 7 (5,8) | 73 (68,79) |
| **Sex** *N (%)* | | | |
| Male | 36,171 (51.4%) | 119,400 (51.2%) | 214,009 (47.1%) |
| Female | 34,244 (48.6%) | 113,872 (48.8%) | 240,775 (52.9%) |
| **Income deprivation decile[2]** *N (%)* | | | |
| D1 (Most deprived) | 16,612 (23.6%) | 55,478 (23.8%) | 66,568 (14.6%) |
| D2 | 13,180 (18.7%) | 43,501 (18.6%) | 60,309 (13.3%) |
| D3 | 8,281 (11.8%) | 27,104 (11.6%) | 45,252 (10.0%) |
| D4 | 6,539 (9.3%) | 20,850 (8.9%) | 40,744 (9.0%) |
| D5 | 4,732 (6.7%) | 15,019 (6.4%) | 41,541 (9.1%) |
| D6 | 3,696 (5.2%) | 12,178 (5.2%) | 40,447 (8.9%) |
| D7 | 3,933 (5.6%) | 12,917 (5.5%) | 46,625 (10.3%) |
| D8 | 3,936 (5.6%) | 13,253 (5.7%) | 44,664 (9.8%) |
| D9 | 4,146 (5.9%) | 14,412 (6.2%) | 40,415 (8.9%) |
| D10 (Least deprived) | 5,364 (7.6%) | 18,565 (8.0%) | 28,222 (6.2%) |
| **Received flu vaccine (2021/22 season)** *N (%)* | | | |
| No | 45,844 (65.1%) | 114,054 (48.9%) | 123,695 (27.2%) |
| Yes | 24,575 (34.9%) | 119,223 (51.1%) | 331,092 (72.8%) |
| **Died during follow-up** *N (%)* | | | |
| No | 70,408 (100.0%) | 233,261 (100.0%) | 440,778 (96.9%) |
| Yes | 11 (<0.1%) | 16 (<0.1%) | 14,009 (3.1%) |
| **Greater Manchester locality** *N (%)* | | | |
| Manchester | 14,574 (20.7%) | 49,917 (21.4%) | 59,746 (13.1%) |
| Oldham | 6,455 (9.2%) | 20,413 (8.8%) | 34,478 (7.6%) |
| Bolton | 7,496 (10.6%) | 25,116 (10.8%) | 50,574 (11.1%) |
| Salford | 7,330 (10.4%) | 22,022 (9.4%) | 34,172 (7.5%) |
| Rochdale | 5,909 (8.4%) | 19,540 (8.4%) | 37,729 (8.3%) |
| Tameside | 5,011 (7.1%) | 16,061 (6.9%) | 34,692 (7.6%) |
| Wigan | 6,732 (9.6%) | 21,424 (9.2%) | 60,712 (13.3%) |
| Bury | 4,583 (6.5%) | 15,157 (6.5%) | 36,297 (8.0%) |
| Trafford | 5,605 (8.0%) | 20,078 (8.6%) | 42,544 (9.4%) |
| Stockport | 6,724 (9.5%) | 23,549 (10.1%) | 63,843 (14.0%) |

[1]IQR, interquartile range.

[2]IDACI (income deprivation affecting children index) used as income deprivation measure for age 2–3 years and age 4–9 years; IDAOPI (income deprivation affecting older people index) used as income deprivation measure for age 65 plus years.

[0.81,0.84])) (Fig 2, S1, S3, and S5 Tables). For comparison, estimates unadjusted by age are shown in S2, S4, and S6 Tables.

There was no clear prepandemic trend in the extent of uptake inequalities over time among preschool children. However, inequalities increased moderately after the onset of the pandemic to HR 0.47 (95% CI [0.45,0.50]) in 2020/21 and HR 0.49 (95% CI [0.47,0.51]) in 2021/22 (Fig 2, S1 Table).

Among primary school children, although flu vaccine uptake inequalities between the most and least income-deprived were relatively stable in the 2 prepandemic seasons, across most deciles inequalities tended to increase (Fig 2, S3 Table). Following the onset of the pandemic,

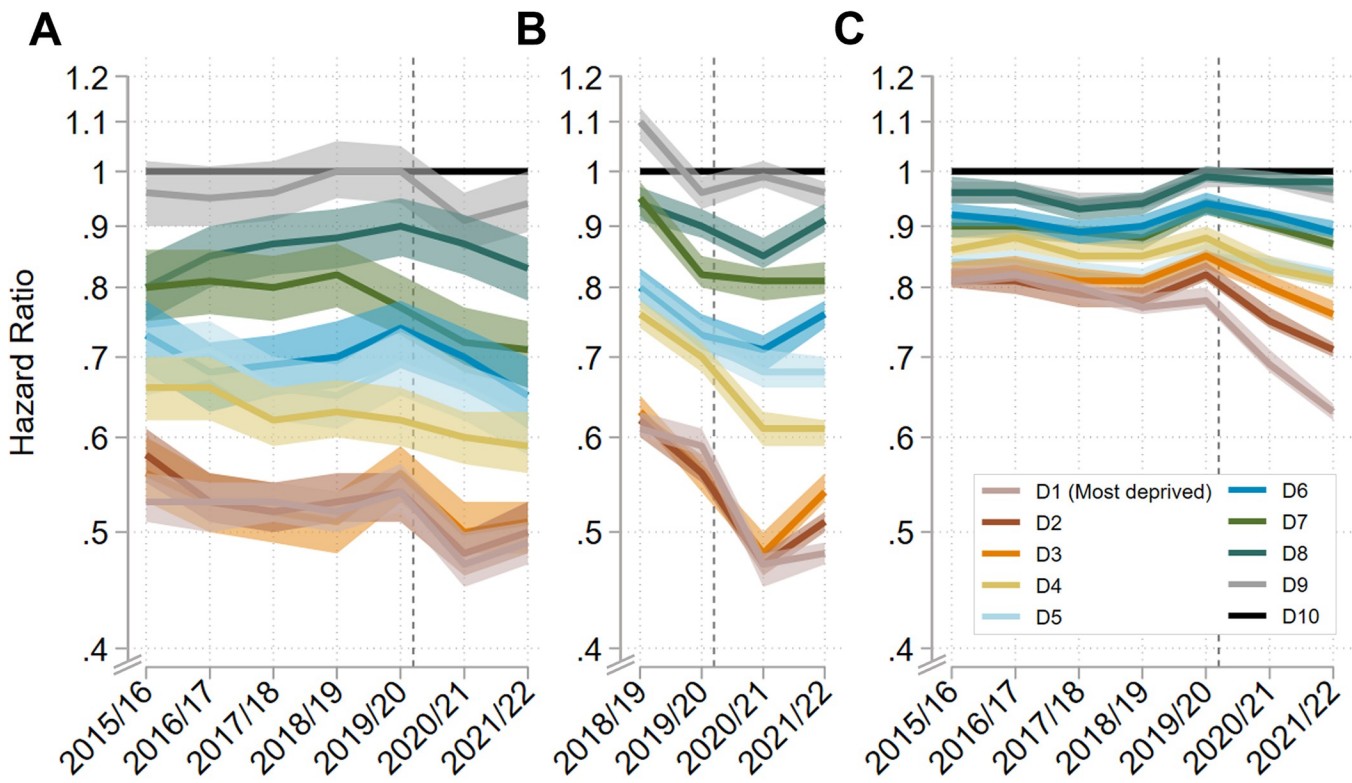

D: income deprivation decile (D1 = most deprived)

**Fig 2. Relative age-adjusted income deprivation-related inequalities in flu vaccine uptake. Results from Cox proportional hazards models adjusted by age are plotted as HRs with 95% CIs.** The reference group is D10 (least deprived areas) for each season. The vertical dashed grey line indicates the onset of the pandemic. (**A**) Inequalities among preschool children (age 2–3 years), estimated using the IDACI. (**B**) Inequalities among primary school children (age 4–9 years), estimated using IDACI (**C**) Inequalities among older adults (age 65 years plus), estimated using IDAOPI. CI, confidence interval; HR, hazard ratio; IDACI, income deprivation affecting children index; IDAOPI, income deprivation affecting older people index.

inequalities between the most and least income-deprived increased substantially, now equivalent to the inequalities among preschool children (HR 0.47 (95% CI [0.45,0.48]) in 2020/21 and HR 0.48 (95% CI [0.47,0.49]) in 2021/22).

Relative income-related inequalities in flu vaccine uptake among older adults increased slightly over the prepandemic flu vaccination seasons, with uptake in the most deprived compared to the least deprived decile ranging from HR 0.81 (95% CI [0.80,0.83]) in 2015/16 to HR 0.78 (95% CI [0.77,0.70]) in 2019/20. This was followed by sharper increases in inequality after the onset of the COVID-19 pandemic, with uptake in the most deprived compared to the least deprived decile reaching HR 0.69 (95% CI [0.68,0.71]) in 2020/21 and HR 0.63 (95% CI [0.62,0.64]) in 2021/22 (Fig 2, S5 Table).

Results were robust to excluding individuals who died during the study period (S7, S8, and S9 Tables) and were generally robust to using total IMD as an alternative deprivation measure (S10, S11, and S12 Tables), though estimated changes in inequalities were slightly smaller among older adults when using total IMD (S12 Table). Results for primary school children and older adults were robust to excluding those on the border of age eligibility (S14 and S15 Tables). However, among preschool children, excluding those who may have attended primary

school resulted in estimates of inequality that were no longer statistically significantly different across vaccination seasons (S13 Table). Estimated inequalities in 2021/22 were slightly wider following inclusion of those eligible due to expanded 2021/22 age criteria (HR most deprived 0.54 (95% CI [0.54,0.55])) and people clinically eligible for flu vaccination (HR 0.56 most deprived (95% CI [0.55,0.56])) compared to the restricted main study population (HR most deprived 0.57 (95% CI [0.56,0.58])) (S16 Table). Stratification by sex (male or female) indicated that socioeconomic inequalities in flu vaccine uptake did not vary substantially by sex across any age group or vaccination season (S17–S22 Tables).

## Absolute inequalities

During prepandemic flu vaccination seasons, age-standardised flu vaccine uptake among each age group tended to gradually increase each season among people living in the least income-deprived neighbourhoods as well as among those in the most income-deprived areas (Fig 3 and S23 Table).

Among preschool children, the magnitude of increases in uptake tended to be slightly higher in the least deprived areas, such that the large uptake gap between the most and least

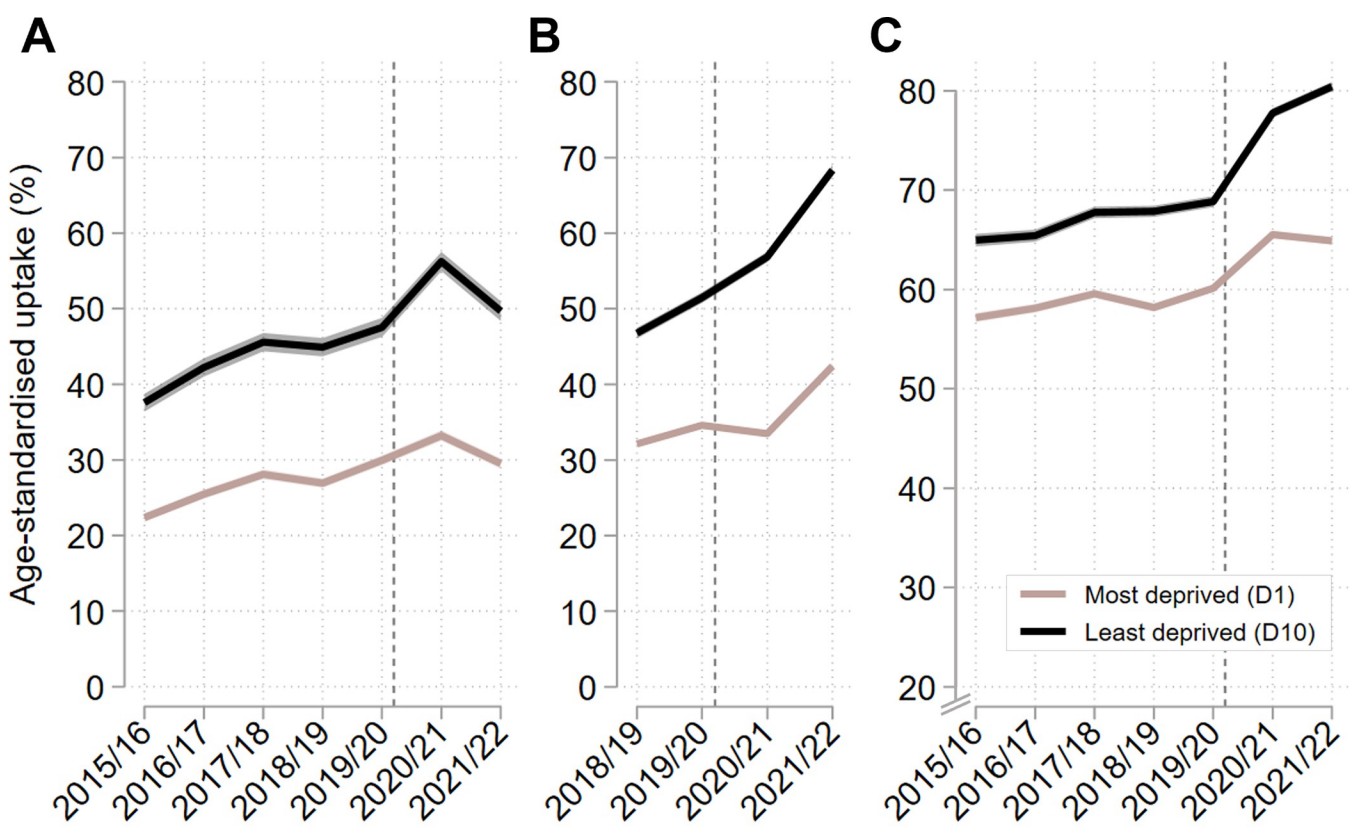

**Fig 3. Absolute age-adjusted income deprivation-related inequalities in flu vaccine uptake over time.** Age-standardised vaccine uptake (%) with 95% CIs. The vertical dashed grey line indicates the onset of the pandemic. (**A**) Inequalities among preschool children (age 2–3 years), estimated using the IDACI. (**B**) Inequalities among primary school children (age 4–9 years), estimated using IDACI. (**C**) Inequalities among older adults (age 65 years plus), estimated using IDAOPI. CI, confidence interval; IDACI, income deprivation affecting children index; IDAOPI, income deprivation affecting older people index.

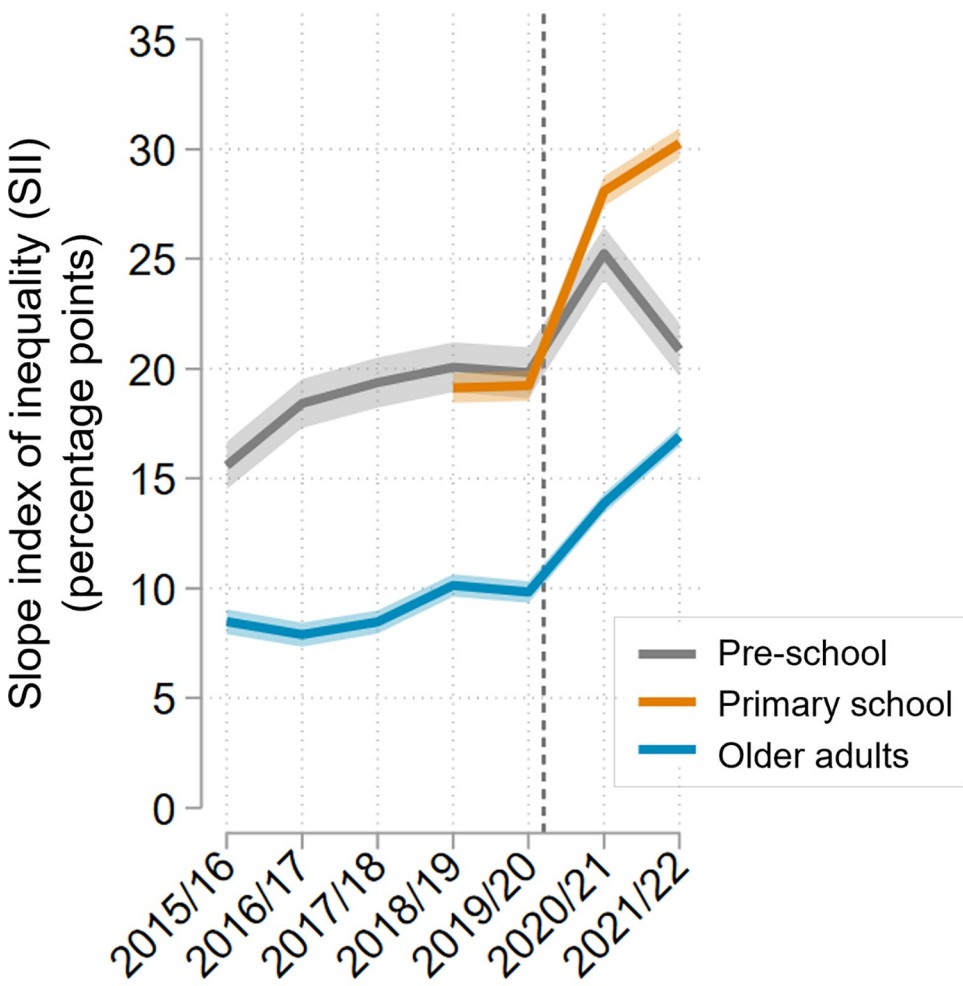

**Fig 4. SII by income deprivation for age-adjusted flu vaccine uptake inequalities over time.** Age-adjusted estimated difference in vaccine uptake (percentage points) between the least and most income-deprived areas. Estimates shown with 95% CIs. The vertical dashed grey line indicates the onset of the pandemic. Results for preschool (age 2–3 years) and primary school (age 4–9 years) children estimated using IDACI; results for older adults (age 65 years plus) estimated using IDAOPI. CI, confidence interval; IDACI, income deprivation affecting children index; IDAOPI, income deprivation affecting older people index; SII, slope index of inequality.

deprived (estimated as the SII) increased from 15.59 percentage points (95% CI [14.52,16.67]) in 2015/16 to 19.82 (95% CI [18.66,20.98]) percentage points in 2019/20. Following the onset of the pandemic, the uptake gap increased more sharply to 25.25 (95% CI [24.04,26.45]) percentage points in 2020/21, driven by disproportionately higher uptake among people in the least deprived areas. The following season (2021/22), flu vaccine uptake among preschool children returned to approximately prepandemic levels, with the uptake gap also reducing to 20.86 (95% CI [19.65,22.05]) percentage points (Figs 3 and 4 and S23 and S24 Tables).

Among primary school children, the prepandemic vaccine uptake gap was similarly large (19.12 (95% CI [19.65,22.05]) to 19.23 (95% CI [18.54,19.91]) percentage points). Following the onset of the pandemic, there were disproportionately higher increases in vaccine uptake among those in the least deprived areas, leading to a widening of the uptake gap to 28.09 (95%

CI [27.41,28.77]) percentage points in 2020/21 and further to 30.27 (95% CI [29.58,30.95]) percentage points in 2021/22 (Figs 3 and 4 and S23 and S24 Tables).

Absolute inequalities in flu vaccine uptake were smaller among older adults in 2015/16, with an uptake gap of 8.48 (95% CI [7.91,9.04]) percentage points between the least and most income deprived. This uptake gap doubled over the study period to 16.91 (95% CI [16.46,17.36]) percentage points in 2021/22, with most of this increase (approximately 80%) occurring since the onset of the COVID-19 pandemic (Fig 4 and S24 Table). Again, widening inequalities were driven primarily by increases in uptake among those in the least deprived areas, with uptake in the most deprived areas increasing or stable in most years (Fig 3 and S23 Table).

## Discussion

Income deprivation was associated with lower flu vaccine uptake among children and older adults in GM prior to the pandemic. Following the onset of the COVID-19 pandemic, flu vaccine uptake increased across all age groups, but uptake increased disproportionately in the least deprived areas, widening preexisting inequalities. Both overall uptake and uptake inequalities increased further among primary school children and older adults in 2021/22, while among preschool children both overall uptake and uptake inequalities reverted to close to prepandemic levels. Although vaccine uptake in the most deprived areas increased for each age group across the study period, 2021/22 age-standardised uptake in deprived areas remained below, or similar to, 2015/16 uptake levels in the least deprived areas. Meanwhile, in the least deprived areas, age-standardised uptake climbed 12 to 22 percentage points over the study period.

The moderate to large prepandemic socioeconomic inequalities in flu vaccine uptake are consistent with previous reports [4,5]. Existing research has suggested that these inequalities may be partly explained by differences across socioeconomic groups in health knowledge about the potential consequences of flu infection and about vaccine effectiveness and safety [5]. Previous studies also found that although flu vaccine uptake is usually highly correlated with previous flu vaccine uptake [28], the COVID-19 pandemic was associated with increases in people intending to take up vaccination for the first time [14]. In a large survey, the most frequently reported motivations for first-time flu vaccine uptake were to protect oneself—both from flu directly and from coinfection with Severe Acute Respiratory Syndrome Coronavirus 2 (SARS-CoV-2)—as well as uptake being viewed as "sensible" and a way to protect healthcare capacity during pandemic pressures [14]. Given that older adults are more vulnerable than children to severe COVID-19 [29], these changes in vaccine uptake intention may apply more to the older adult population, potentially explaining the wider increases in inequalities among this group. Conversely, the lower risk of severe COVID-19 in children led to substantial debate about the relative risks and benefits of vaccinating children and adolescents against COVID-19 [30,31], with the UK being slower than some other high-income countries to offer vaccination to those under 18 years [32]. These policy debates may have highlighted the potential risks of vaccination to parents, potentially also increasing parental uncertainty about other vaccines during childhood.

Our estimates of overall flu vaccine uptake within GM are consistently lower than national uptake estimates [6,33], likely in part due to above-average income deprivation within the region. Estimated uptake was particularly low among preschool children (34.9% in 2021/22), which was also substantially lower than estimated national uptake (50.1%) [33]. However, our estimates of uptake among this age group are consistent with published data from Merseyside, a similarly deprived area in the North West of England, where almost 80% of neighbourhoods

had less than 35% flu vaccine uptake among preschool children in 2015/16 [5]. The authors suggest that this may be in part due to lack of parental awareness of the risks of flu infection to young children, despite relatively high rates of hospitalisations among this age group [5]. However, low estimated uptake among children may also be partly driven by issues with assigning eligibility based on age at the index date discussed in more detail below, and we cannot exclude the possibility of some missing vaccination records in our dataset. Although this could have affected exact estimates of inequalities, given the consistent differences in uptake estimates over time, it seems unlikely that this could have driven our finding that inequalities increased.

A strength of this study is the use of a dataset with almost population-level coverage for a large city region [24], along with robust measures of income deprivation that specifically capture the proportion of children or older adults living in poverty at a neighbourhood level [21]. The availability of data covering 7 vaccination seasons also allows us to robustly identify changes during the pandemic. A limitation of this analysis is the focus on a single region (GM), which may limit generalisability to other geographical contexts. However, GM is a large ethnically and socioeconomically diverse metropolitan region, which also includes smaller towns and some rural areas [20,21,24]. We therefore think the observed widening of inequalities is likely to apply more broadly to the UK population.

Another limitation of the dataset is that age is included only as age in years at the index date (1 February 2020), to minimise any potentially identifiable patient information. While this allowed us to calculate age in years on 1 February, for each vaccination season, it will have introduced some errors in assigned flu vaccine eligibility. Sensitivity analysis indicated that this is unlikely to have affected the results for primary school children or older adults. However, the widening inequalities among preschool children may have been partially driven by the inclusion of some school-age children in this group.

There is some survival bias in this dataset, as some people who died before the index date (1 February 2020) are excluded. While this could introduce bias, sensitivity analysis excluding those who died at any point in the study period indicated that this is unlikely to have substantially affected estimates of inequality. Another data limitation is that we were unable to reliably identify people who were eligible for flu vaccination due to health conditions over time. Sensitivity analysis including current clinical eligibility and expanded age eligibility for the 2021/22 season suggested that our results likely slightly underestimate the full extent of uptake inequalities across the total population.

Although we did not find evidence that socioeconomic inequalities in flu vaccine uptake varied by sex across any age group, there are likely other intersecting axes of health inequality that may contribute to overall inequalities in flu vaccine uptake. For example, a limitation of our analysis is that we did not explore inequalities by ethnic group, which is a known determinant of both flu vaccine and COVID-19 vaccine uptake in this study population [24]. Given that structural racism results in people from ethnic minority backgrounds being overrepresented in more deprived areas [34], our results raise the possibility that ethnic inequalities in flu vaccine uptake may also have widened, which will be an important area for future research.

This study, together with previous evidence, indicates that the COVID-19 pandemic was associated with increases in flu vaccine uptake across all eligible age group and all levels of deprivation. However, uptake increased disproportionately in less income-deprived areas, potentially due in part to inequalities in access to vaccination. The COVID-19 pandemic was associated with increased demand for primary care, alongside reduced service capacity and additional primary care workload due to COVID-19 vaccine delivery [9]. While these factors likely impacted all areas, there are fewer GPs per population in more deprived areas [35], so deprived areas likely experienced increased disruptions in access to routine care.

Similarly, there was widespread disruption to in-person schooling, but children in deprived areas experienced greater loss of in-school time due to longer periods of time subject to government-mandated school closures as well as increased staff and pupil absences resulting from COVID-19 sickness or requirements to self-isolate [10]. This may have contributed to increased inequalities in flu vaccine uptake among school-aged children.

Another factor that may have affected inequalities in access to flu vaccination is the differential impact of long-term ill health due to long COVID [36] and pandemic-related delays in elective healthcare [37]. While increased morbidity may have increased some people's perceived need for flu vaccination, worsening health might also have increased the difficulty of accessing vaccination. Given the higher burden of both long COVID [36] and long-term health conditions [38] among people living in more deprived areas, these factors could have contributed to widening inequalities.

Finally, the COVID-19 pandemic may also have differentially affected demand for flu vaccination between more and less deprived areas. Overall increases in vaccine uptake suggest that public health messaging and increased awareness of the personal and societal importance of flu vaccination [6,14,28] may have increased demand for flu vaccination. However, misinformation and disinformation shared on social media is also linked to vaccine uptake intentions, and there were large volumes of anti-vaccine disinformation associated with the COVID-19 vaccines [13–15]. Similarly, the pandemic was associated with extensive sharing of false conspiracy theories related to the COVID-19 vaccines and the pandemic more broadly, and endorsement of such ideas has been shown to be associated with lower intention to take up vaccination [39]. Trust in public health messaging over misinformation and disinformation is therefore important for vaccine uptake, and there may have been social gradients in exposure to, or responses to, recent pro- and anti-vaccine messaging.

Some of the greater increases in demand for flu vaccination among older adults may also be linked to the option for coadministration with COVID-19 vaccines, perhaps due to convenience, or to greater perception of risk from COVID-19. However, given the steep socioeconomic gradients in uptake of COVID-19 vaccination [40], this may also have inadvertently contributed to wider flu vaccine uptake inequalities.

Comparing between age groups, inequalities were widest among primary school children by the end of the study period. This is interesting, as delivery within schools could reduce barriers to vaccination compared to the preschool group, for whom a GP appointment is required. The overall higher rates of uptake among school children compared to preschool children suggest that the school setting may indeed facilitate uptake, perhaps especially in the second pandemic season where uptake among preschool children fell across all deprivation levels. However, more research is required to understand the much wider inequalities within the school context since the onset of the COVID-19 pandemic.

Although flu circulation and disease were very low during 2020/21 and 2021/22, this was likely driven by COVID-19 control measures [41]. However, most COVID-19 control measures have since been suspended, and winter 2022/23 has seen high levels of SARS-CoV-2 and flu virus cocirculation and increased flu-related hospitalisations [42]. Alongside lower flu vaccine uptake, people in more deprived areas are also less likely to have taken up COVID-19 vaccination and are more likely to have long-term health conditions that increase the risk of severe outcomes if infected [38,40]. Wider inequalities in flu vaccine uptake are therefore likely to translate into increased morbidity and mortality inequalities over time.

Here, we report widening income-related inequalities in routine flu vaccination among children and older adults in the context of the COVID-19 pandemic. Further research is needed to understand whether inequalities along other sociodemographic axes of inequality and for other routine vaccinations have also widened. Finally, it is crucial to improve our

understanding of what has driven the increase in vaccine uptake inequalities and to address and reverse these growing inequalities.

## Supporting information

**S1 Fig. Study population flow chart.**
(DOCX)

**S2 Fig. Log(−log[survival]) versus log(time) plots by deprivation decile for Cox proportional hazards models by age group and vaccination season.**
(DOCX)

**S1 RECORD. Checklist of items, extended from the STROBE statement, which should be reported in observational studies using routinely collected health data.**
(DOCX)

**S1 Table. Relative age-adjusted income deprivation-related inequalities in flu vaccine uptake among preschool children (age 2–3 years).** Results from Cox proportional hazards models adjusted by age are reported as hazard ratios with 95% confidence intervals. The reference groups are D10 (least deprived areas) and age 2 years for each season. The vertical line indicates the onset of the pandemic. Results also shown in Fig 2 in the main text.
(DOCX)

**S2 Table. Relative unadjusted income deprivation-related inequalities in flu vaccine uptake among preschool children (age 2–3 years) (for comparison with S1 Table).** Results from Cox proportional hazards models are reported as hazard ratios with 95% confidence intervals. The reference group is D10 (least deprived areas) for each season. The vertical line indicates the onset of the pandemic.
(DOCX)

**S3 Table. Relative age-adjusted income deprivation-related inequalities in flu vaccine uptake among primary school children (age 4–9 years).** Results from Cox proportional hazards models adjusted by age are reported as hazard ratios with 95% confidence intervals. The reference groups are D10 (least deprived areas) and age 4 years for each season. The vertical line indicates the onset of the pandemic. Results also shown in Fig 2 in the main text.
(DOCX)

**S4 Table. Relative unadjusted income deprivation-related inequalities in flu vaccine uptake among primary school children (age 4–9 years) (for comparison with S3 Table).** Results from Cox proportional hazards models are reported as hazard ratios with 95% confidence intervals. The reference group is D10 (least deprived areas). The vertical line indicates the onset of the pandemic.
(DOCX)

**S5 Table. Relative age-adjusted income deprivation-related inequalities in flu vaccine uptake among older adults (age 65 years plus).** Results from Cox proportional hazards models adjusted by age are reported as hazard ratios with 95% confidence intervals. The reference groups are D10 (least deprived areas) and age 65–69 years for each season. The vertical line indicates the onset of the pandemic. Results also shown in Fig 2 in the main text.
(DOCX)

**S6 Table. Relative unadjusted income deprivation-related inequalities in flu vaccine uptake among older adults (age 65 years plus) (for comparison with S5 Table).** Results from Cox proportional hazards models are reported as hazard ratios with 95% confidence

intervals. The reference group is D10 (least deprived areas) for each season. The vertical line indicates the onset of the pandemic.
(DOCX)

**S7 Table. Relative age-adjusted income deprivation-related inequalities in flu vaccine uptake among preschool children (age 2–3 years)—Sensitivity analysis excluding individuals who died during the study period.** Results from Cox proportional hazards models adjusted by age are reported as hazard ratios with 95% confidence intervals. The reference groups are D10 (least deprived areas) and age 2 years for each season. The vertical line indicates the onset of the pandemic.
(DOCX)

**S8 Table. Relative age-adjusted income deprivation-related inequalities in flu vaccine uptake among primary school children (age 4–9 years)—Sensitivity analysis excluding individuals who died during the study period.** Results from Cox proportional hazards models adjusted by age are reported as hazard ratios with 95% confidence intervals. The reference groups are D10 (least deprived areas) and age 4 years for each season. The vertical line indicates the onset of the pandemic.
(DOCX)

**S9 Table. Relative age-adjusted income deprivation-related inequalities in flu vaccine uptake among older adults (age 65 years plus)–Sensitivity analysis excluding individuals who died during the study period.** Results from Cox proportional hazards models adjusted by age are reported as hazard ratios with 95% confidence intervals. The reference groups are D10 (least deprived areas) and age 65–69 years for each season. The vertical line indicates the onset of the pandemic.
(DOCX)

**S10 Table. Relative age-adjusted multiple deprivation-related inequalities in flu vaccine uptake among preschool children (age 2–3 years)—Sensitivity analysis using the index of multiple deprivation (IMD) as an alternative measure of deprivation.** Results from Cox proportional hazards models adjusted by age are reported as hazard ratios with 95% confidence intervals. The reference groups are D10 (least deprived areas) and age 2 years for each season. The vertical line indicates the onset of the pandemic.
(DOCX)

**S11 Table. Relative age-adjusted multiple deprivation-related inequalities in flu vaccine uptake among primary school children (age 4–9 years)—Sensitivity analysis using the index of multiple deprivation (IMD) as an alternative measure of deprivation.** Results from Cox proportional hazards models adjusted by age are reported as hazard ratios with 95% confidence intervals. The reference groups are D10 (least deprived areas) and age 4 years for each season. The vertical line indicates the onset of the pandemic.
(DOCX)

**S12 Table. Relative age-adjusted multiple deprivation-related inequalities in flu vaccine uptake among older adults (age 65 years plus)—Sensitivity analysis using the index of multiple deprivation (IMD) as an alternative measure of deprivation.** Results from Cox proportional hazards models adjusted by age are reported as hazard ratios with 95% confidence intervals. The reference groups are D10 (least deprived areas) and age 65–69 years for each season. The vertical line indicates the onset of the pandemic.
(DOCX)

**S13 Table. Relative age-adjusted income deprivation-related inequalities in flu vaccine uptake among preschool children (age 2–3 years)—Sensitivity analysis excluding children on the border of age-based vaccine eligibility (i.e., excluding age 3/4 years).** Results from Cox proportional hazards models adjusted by age are reported as hazard ratios with 95% confidence intervals. The reference groups are D10 (least deprived areas) and age 2 years for each season. The vertical line indicates the onset of the pandemic.
(DOCX)

**S14 Table. Relative age-adjusted income deprivation-related inequalities in flu vaccine uptake among primary school children (age 4–9 years)—Sensitivity analysis excluding children on the border of age-based vaccine eligibility (i.e., excluding age 9/10 years).** Results from Cox proportional hazards models adjusted by age are reported as hazard ratios with 95% confidence intervals. The reference groups are D10 (least deprived areas) and age 4 years for each season. The vertical line indicates the onset of the pandemic.
(DOCX)

**S15 Table. Relative age-adjusted income deprivation-related inequalities in flu vaccine uptake among older adults (age 65 years plus)—Sensitivity analysis excluding adults on the border of age-based vaccine eligibility (i.e., excluding age 64/65 years).** Results from Cox proportional hazards models adjusted by age are reported as hazard ratios with 95% confidence intervals. The reference groups are D10 (least deprived areas) and age 66–69 years for each season. The vertical line indicates the onset of the pandemic.
(DOCX)

**S16 Table. Relative age-adjusted deprivation-related inequalities in flu vaccine uptake for 2021/22 vaccination season—Sensitivity analysis comparing all-age inequalities across (1) main sample (2) expanded age eligibility for 2021/22 and (3) expanded age eligibility and clinical eligibility.** Results from Cox proportional hazards models adjusted by age are reported as hazard ratios with 95% confidence intervals. The reference groups are D10 (least deprived areas), age 0–4 years, and no clinical eligibility for flu vaccination for each season. Deprivation measure is the index of multiple deprivation (IMD).
(DOCX)

**S17 Table. Relative age-adjusted income deprivation-related inequalities in flu vaccine uptake among preschool children (age 2–3 years) stratified by sex—Male results.** Results from Cox proportional hazards models adjusted by age are reported as hazard ratios with 95% confidence intervals. The reference groups are D10 (least deprived areas) and age 2 years for each season. The vertical line indicates the onset of the pandemic.
(DOCX)

**S18 Table. Relative age-adjusted income deprivation-related inequalities in flu vaccine uptake among preschool children (age 2–3 years) stratified by sex—Female results.** Results from Cox proportional hazards models adjusted by age are reported as hazard ratios with 95% confidence intervals. The reference groups are D10 (least deprived areas) and age 2 years for each season. The vertical line indicates the onset of the pandemic.
(DOCX)

**S19 Table. Relative age-adjusted income deprivation-related inequalities in flu vaccine uptake among primary school children (age 4–9 years) stratified by sex—Male results.** Results from Cox proportional hazards models adjusted by age are reported as hazard ratios with 95% confidence intervals. The reference groups are D10 (least deprived areas) and age 4

years for each season. The vertical line indicates the onset of the pandemic.
(DOCX)

**S20 Table. Relative age-adjusted income deprivation-related inequalities in flu vaccine uptake among primary school children (age 4–9 years) stratified by sex—Female results.** Results from Cox proportional hazards models adjusted by age are reported as hazard ratios with 95% confidence intervals. The reference groups are D10 (least deprived areas) and age 4 years for each season. The vertical line indicates the onset of the pandemic.
(DOCX)

**S21 Table. Relative age-adjusted income deprivation-related inequalities in flu vaccine uptake among older adults (age 65 years plus) stratified by sex—Male results.** Results from Cox proportional hazards models adjusted by age are reported as hazard ratios with 95% confidence intervals. The reference groups are D10 (least deprived areas) and age 65–69 years for each season. The vertical line indicates the onset of the pandemic.
(DOCX)

**S22 Table. Relative age-adjusted income deprivation-related inequalities in flu vaccine uptake among older adults (age 65 years plus) stratified by sex—Female results.** Results from Cox proportional hazards models adjusted by age are reported as hazard ratios with 95% confidence intervals. The reference groups are D10 (least deprived areas) and age 65–69 years for each season. The vertical line indicates the onset of the pandemic.
(DOCX)

**S23 Table. Absolute age-adjusted income deprivation-related inequalities in flu vaccine uptake over time.** Age-standardised vaccine uptake (%) with 95% confidence intervals. The vertical line indicates the onset of the pandemic. Results also shown in Fig 3 in the main text.
(DOCX)

**S24 Table. Slope index of inequality (SII) by income deprivation for age-adjusted flu vaccine uptake inequalities over time.** Age-adjusted estimated difference in vaccine uptake (percentage points) between the least and most income-deprived areas. Estimates shown with 95% confidence intervals. The vertical line indicates the onset of the pandemic. Results also shown in Fig 4 in the main text.
(DOCX)

## Acknowledgments

We are grateful to Nicky Timmis, Aneela McAvoy, Joanna Ferguson, and Sue Wood for their support in organising and hosting PCIE discussion groups. We would like to thank all members of the NIHR Applied Research Collaboration for Greater Manchester Public and Community Involvement and Engagement Panel and the Health Innovation Manchester Public Community Involvement and Engagement Forum. We would particularly like to thank Nasrine Akhtar, Basma Issa, Nicholas Filer, and Charles Kwaku-Odoi for all their input as part of the Vaccine Equity Project Public Advisory Group. We would also like to recognise the GMCR (a partnership of Greater Manchester Health and Social Care Partnership, Health Innovation Manchester, and Graphnet Health, on behalf of Greater Manchester localities) for the provision of data required to undertake this work. This work uses data provided by patients and collected by the NHS as part of patient care and support. Using patient data is vital to improving health and care for everyone. There is huge potential to make better use of information from people's patient records, to understand more about disease, develop new

treatments, monitor safety, and plan NHS services. Patient data should be kept safe and secure, to protect everyone's privacy, and it is important that there are safeguards to make sure that data are stored and used responsibly. Everyone should be able to find out about how patient data are used.

## Author Contributions

**Conceptualization:** Ruth Elizabeth Watkinson, Stephanie Gillibrand, Luke Munford, Matt Sutton.

**Data curation:** Richard Williams.

**Formal analysis:** Ruth Elizabeth Watkinson.

**Funding acquisition:** Matt Sutton.

**Investigation:** Ruth Elizabeth Watkinson, Stephanie Gillibrand, Luke Munford, Matt Sutton.

**Methodology:** Ruth Elizabeth Watkinson, Richard Williams, Stephanie Gillibrand, Matt Sutton.

**Project administration:** Ruth Elizabeth Watkinson.

**Resources:** Matt Sutton.

**Supervision:** Luke Munford, Matt Sutton.

**Validation:** Richard Williams.

**Visualization:** Ruth Elizabeth Watkinson.

**Writing – original draft:** Ruth Elizabeth Watkinson.

**Writing – review & editing:** Ruth Elizabeth Watkinson, Richard Williams, Stephanie Gillibrand, Luke Munford, Matt Sutton.

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
