## [Editor Report · Decision Letter 0]

17 Feb 2023

Dear Dr Watkinson, 

Thank you for submitting your manuscript entitled "Widening socioeconomic inequalities in flu vaccine uptake during the COVID-19 pandemic : a cohort study in Greater Manchester, England" for consideration by PLOS Medicine.

Your manuscript has now been evaluated by the PLOS Medicine editorial staff as well as by an academic editor with relevant expertise and I am writing to let you know that we would like to send your submission out for external peer review.

Please re-submit your manuscript within two working days, i.e. by Feb 21 2023 11:59PM.

Kind regards,

Callam Davidson

Senior Editor

PLOS Medicine

---

## [Decision Letter · Decision Letter 1]

9 May 2023

Dear Dr. Watkinson,

Thank you very much for submitting your manuscript "Widening socioeconomic inequalities in flu vaccine uptake during the COVID-19 pandemic : a cohort study in Greater Manchester, England" (PMEDICINE-D-23-00344R1) for consideration at PLOS Medicine. 

Your paper was evaluated by an associate editor and discussed among all the editors here. It was also discussed with an academic editor with relevant expertise, and sent to independent reviewers, including a statistical reviewer. The reviews are appended at the bottom of this email and any accompanying reviewer attachments can be seen via the link below:

[LINK]

In light of these reviews, I am afraid that we will not be able to accept the manuscript for publication in the journal in its current form, but we would like to consider a revised version that addresses the reviewers' and editors' comments. Obviously we cannot make any decision about publication until we have seen the revised manuscript and your response, and we plan to seek re-review by one or more of the reviewers. 

We expect to receive your revised manuscript by May 30 2023 11:59PM. Please email us (plosmedicine@plos.org) if you have any questions or concerns.

We look forward to receiving your revised manuscript. 

Sincerely,

Alexandra Schaefer, PhD

PLOS Medicine

plosmedicine.org

GENERAL

Please respond to all editor and reviewer comments detailed below in full.

Please ensure consistency in your number format and revise your manuscript accordingly including the Supplementary Material (51·4% or 51.4%).

For all observational studies, in the manuscript text, please indicate: (1) the specific hypotheses you intended to test, (2) the analytical methods by which you planned to test them, (3) the analyses you actually performed, and (4) when reported analyses differ from those that were planned, transparent explanations for differences that affect the reliability of the study's results. If a reported analysis was performed based on an interesting but unanticipated pattern in the data, please be clear that the analysis was data-driven.

Did your study have a prospective protocol or analysis plan? Please state this (either way) early in the Methods section.

FINACIAL DISCLOSURE

It appears that one or more study authors is affiliated with one or more of the agencies that funded the study. Thus, the statement “The funders had no role in study design, data collection and analysis, decision to publish, or preparation of the manuscript” does not apply. Please revise the Financial Disclosure accordingly, as in "[Author name] is [author's role] at [funding agency]. The funders had no other role in study design…..”

COMPETING INTEREST STATEMENT

All authors must declare their relevant competing interests per the PLOS policy, which can be seen here:

https://journals.plos.org/plosmedicine/s/competing-interests

For authors with ties to industry, please indicate whether any of the interests has a financial stake in the results of the current study.

ABSTRACT 

PLOS Medicine requests that main results are quantified with 95% CIs as well as p values. Please include. When reporting p values please report as p<0.001 and where higher as the exact p value p=0.002, for example. For the purposes of transparent data reporting, if not including the aforementioned please clearly state the reasons why not.

Please include any important dependent variables that are adjusted for in the analyses.

Abstract Background: Provide the context of why the study is important. The final sentence should clearly state the study question.

Abstract Methods and Findings:

* Please include the number of participants.

Please define all abbreviations used for statistical reporting at first use.

Line 37 – Please define NHS at first in the abstract. 

Line 42 – Throughout, suggest reporting statistical information as follows to improve clarity for the reader “22% (95% CI [13%,28%]; p</=)”. Please amend throughout the abstract and main manuscript.

Please note the use of commas to separate upper and lower bounds, as opposed to hyphens as these can be confused with reporting of negative values.

AUTHOR SUMMARY

At this stage, we ask that you include a short, non-technical Author Summary of your research to make findings accessible to a wide audience that includes both scientists and non-scientists. The Author Summary should immediately follow the Abstract in your revised manuscript. This text is subject to editorial change and should be distinct from the scientific abstract. Please see our author guidelines for more information: https://journals.plos.org/plosmedicine/s/revising-your-manuscript#loc-author-summary.

The summary should include 2-3 single sentence, individual bullet points under each of the questions.

It may be helpful to review currently published articles for examples which can be found on our website here https://journals.plos.org/plosmedicine/

INTRODUCTION

Please address the potential importance of your study. Indicate whether your study is novel and how you determined that. If there has been a systematic review of the evidence related to your study (or you have conducted one), please refer to and reference that review and indicate whether it supports the need for your study. 

Considering a global readership, please include a short statement about the role and function of the NHS.

Please elaborate on the choice of greater Manchester as the study setting, the wider sociodemographic and socioeconomic setting associated with the area and provide justification why the study setting is suitable for general validity of the results (outside of Manchester).

METHODS AND RESULTS 

PLOS Medicine requests that the main outcomes are quantified with p values as well as 95% CIs. Please report p values as p<0.001 and where higher as p=0.002, for example. If not including p values, to help facilitate transparent data reporting, please clearly state the reasons why not.

When a p value is given, please specify the statistical test used to determine it.

Please present numerators and denominators for percentages, at least in the Tables.

Suggest reporting statistical information as detailed above – see under ABSTRACT

Line 90 – Please define GP in the Methods.

Line 109 – Please change ‘focus’ to ‘focused’ and ensure to keep the tense consistent throughout the manuscript.

Line 159 – Please define NIHR in the Methods.

Please include in the figure caption of Figure 2 and 3 the age range of pre-school children, primary school children, and older adults.

In Figure 2 and 3, please show the axis beginning at zero. If this is not possible, please show a break in the axis.

In Figure 4, please define g IDACI and IDAOPI and include in the figure caption the age range of pre-school children, primary school children, and older adults.

DISCUSSION

Please present and organize the Discussion as follows: a short, clear summary of the article's findings; what the study adds to existing research and where and why the results may differ from previous research; strengths and limitations of the study; implications and next steps for research, clinical practice, and/or public policy; one-paragraph conclusion.

Lines 308-309: Please include a reference for the following statement “Given that older adults are more vulnerable than children to severe COVID-19, […]”.

FIGURES AND TABLES

Please ensure to define the abbreviations used in your figures and tables 

(e.g. in Table 1 add “IQR = interquartile range” underneath the table)

Please provide titles and legends for all figures (including those in Supporting Information files).

Please indicate in figure captions whether analyses are adjusted or unadjusted and where adjusted please detail the factors adjusted for. 

Where adjusted analyses are presented, to help facilitate transparent data reporting please also unadjusted analyses for comparison.

Where 95% CIs are reported PLOS Medicine also requires that p values are reported. Please include in the tables and figures. Please report p as <0.001 and where higher as p=0.002, for example

Please replace the use of hyphens with commas to separate upper and lower CI bounds as these can be confused with negative values.

Please consider the use of a color palate suitable for those with color blindness.

In the Supplementary Material, in the Table of contents, please change “STROBE Statemen” to “STROBE Statement”.

In the Supplementary Tables, please explain in the caption the values (IDACI decile D10 or age) used as reference points (in the table shown as ‘REF’).

REFERENCES

Please cite your Supporting Information as outlined here: https://journals.plos.org/plosmedicine/s/supporting-information

Please cite the reference numbers in square brackets (e.g., “We used the techniques developed by our colleagues [19] to analyze the data”). Citations should be preceding punctuation.

PLOS uses the numbered citation (citation-sequence) method and first six authors, et al.

Comments from the reviewers:

Reviewer #1: Alex McConnachie, Statistical Review

Watkinson et al present data from a General Practice database covering the Manchester area, looking at socioeconomic differences in flu vaccine uptake from 2015/16 to 2021/22. This review looks at the use of statistics in the paper.

These are generally very good. The data are clearly described and presented. Cox models are used to analyse time to vaccination, and age-standardised vaccination rates are analysed using the slope index of inequality. A number of sensitivity analyses are presented to assess the robustness of the findings. The results appear to be interpreted appropriately. My comments are relatively minor.

In the abstract, the statement "…vaccine uptake increased across all socioeconomic groups during the pandemic" is not supported by any of the data reported at that point.

The section on covariates makes no mention of gender. Looking at Table 1, this data is available, and looks virtually complete. Why was this not considered, at least in the adult cohort? The other obvious factor of interest would be ethnicity, though I assume the data here is less reliable. Should this at least be noted as a limitation?

As ever with Cox models, there is the question of the proportional hazards assumption. Was this checked? My guess, with a dataset of this size, there may be some evidence of non-PH, but as long as it is not too severe, then a PH model should be OK.

In Table 1, age is reported as median and quartiles, which is fine, but why report to one decimal place? I assume the data are integers. Also, on a stylistic note, perhaps the percentages for deaths in the child cohorts could be reported as "<0.1%" and ">99.9%"?

In the discussion, line 342 talks of "this winter". Someone reading this in a few years' time will not immediately know what that means.

Reviewer #2: See attachment

Reviewer #3: Thank you for asking me to review this paper. It is an interesting topic, may help inform future flu vaccination roll-outs and highlights important findings around widening social inequality. 

The abstract the clearly written.

Introduction

Overall clearly written

Line 46 - could you expand here to give some potential cited reasons for lower vaccine uptake in more deprived neighbourhoods? Some general reasons here would be useful for context, before moving onto COVID-19 pandemic specific context. 

Could something be mentioned in the introduction/later in the discussion about the effect of COVID-19 related illness (and long-COVID) and potential influence on flu vaccination uptake?

Methods

Line 124: I would like more information on how this data is recorded/collected.

Results

The results might be better structured with 3 separate subheadings to describe the three very different groups. 

34.9% pre-school uptake seems very low? There are brief reasons described to explain this in the discussion, but this point requires further expansion in the discussion including correlation with previous studies. 

Limitations

A suggested limitation in this study is lack of further information on other socio-demographic factors such as ethnicity. Ethnicity and cultural beliefs have been associated with lower vaccine uptake and higher morbidity/mortality from COVID-19 - it would have been informative to see this data and how it ties in with the widening inequalities by social deprivation, particularly in this urban area of GM

Comparison to literature

This section is very brief.

Line 303 - suggest expand here as to what previous reports have said about pre-pandemic inequalities. 

As described above, need to expand more on previous study findings in relation to each of the three groups (particularly pre-school)

Overall, the conclusion considers a broad range of reasons for widening socio-economic inequalities with flu vaccine uptake during this period. There are 2 points I think should be covered that are missing: 

- Those from more deprived backgrounds experienced much higher rates of COVID-19, morbidity and mortality (including long-Covid) - could this have affected uptake of flu vaccine afterwards? 

- There is some discussion of misinformation, but not much around "vaccine hesitancy" and mistrust, which is prevalent amongst patients from more deprived backgrounds and minority ethnic groups - which was exacerbated during the COVID-19 pandemic; also vaccine hesitancy amongst parents around the time of the COVID-19 vaccine being introduced. Suggested reference for further context: https://journals.plos.org/plosmedicine/article?id=10.1371/journal.pmed.1003826 - though there are others more relevant to this group - but this point is worth expanding on. 

Overall this is a well executed study and clearly written paper with some important points on widening socioeconomic inequalities that would be valuable to those involved in future vaccination planning/ outreach to underserved communities.

[LINK]

---

## [Decision Letter · Decision Letter 2]

23 Jun 2023

Dear Dr. Watkinson,

Thank you very much for re-submitting your manuscript " Widening socioeconomic inequalities in flu vaccine uptake during the COVID-19 pandemic: a cohort study in Greater Manchester, England" (PMEDICINE-D-23-00344R2) for review by PLOS Medicine.

I have discussed the paper with my colleagues and the academic editor and it was also seen again by three reviewers. I am pleased to say that provided the remaining editorial and production issues are dealt with we are planning to accept the paper for publication in the journal.

[LINK]

We look forward to receiving the revised manuscript by Jun 30 2023 11:59PM.   

Sincerely,

Alexandra Schaefer, PhD

Associate Editor 

PLOS Medicine

plosmedicine.org

Requests from Editors:

GENERAL

Thank you for considered and detailed responses to editor and reviewer comments.

Please see below for further minor points that we request you respond to in full.

Please revise your manuscript using either British English or American English, e.g., colored/coloured (Description Figure 1) or age-standardised/age-standardized (Line 267).

Please choose a consistent spelling for the terms “income-deprived”/”income deprived” as well as “income deprivation”/”income-deprivation” and revise you manuscript accordingly.

Please ensure to use a consistent tense within a section of the manuscript.

Please add 'years’ to ages stated e.g. age two to three years, throughout your manuscript 

ACADEMIC EDITOR COMMENTS

The manuscript has much improved and I am happy with the revised version.

TITLE

Please revise your title according to PLOS Medicine's style. Your title must be nondeclarative. We suggest “Evaluating socioeconomic inequalities in influenza vaccine uptake during the COVID-19 pandemic: A cohort study in Greater Manchester, England” or similar

ABSTRACT

Line 33: Please temper assertions of primacy ("no existing studies have investigated…”) by adding ‘to the best of our knowledge’ or similar.

Line 38-9: Please add 'years’ to ages stated e.g., age two to three years.

Lines 43-51: Please add the according unit (e.g., percentage points) to numbers occurring in your abstract.

AUTHOR SUMMARY

The Author Summary should immediately follow the Abstract in your revised manuscript. 

In the third bullet point under ‘Why was this study done?’, please temper assertions of primacy ("no existing studies had analysed…”) by adding ‘to the best of our knowledge’ or similar.

Please revise the second bullet point under “What did the authors do and find?”. Editorial suggestion: “We focused on young children and older adults, as these age groups are at higher risk of severe outcomes from flu infection and are eligible for annual flu vaccination provided free at the point of service.”

In the final bullet point of ‘What Do These Findings Mean?’, please describe the main limitations of the study in non-technical language.

INTRODUCTION

Lines 65-66 suggest: “[…] (young children and adults who are aged over 65, groups with certain preexisting health conditions, or pregnant people [3]).

Lines 74-76 suggest: “Flu vaccine uptake amongst pre-school and school aged children has consistently been lower than amongst older adults, ranging from approximately 30% to 60% pre-pandemic.”

Line 90: Please write million instead of ‘M’.

Lines 93-94 suggest: “[…], despite overall higher levels of deprivation than the national average [21].”

Lines 94-95 suggest: “For these reasons, Greater Manchester is a useful setting to study changes in socioeconomic inequalities in vaccine uptake.”

METHODS

Line 110: Please change “roll-out” to “rollout” since you have used the latter spelling consistently throughout your manuscript.

Line 131: Please define the following abbreviation: UK.

Line 142: Please define the following abbreviation: HM (HM Revenue & Customs)

Please revise the following statement for improved clarity: “In regression analysis age in years was used for children, then age was grouped into five-year age bands up to 80 plus for adults”. Editorial suggestion: “In regression analysis, age in years was used for children and age grouped into five-year age bands up to 80 years plus was used for adults.”

Please ensure that the study is reported according to the RECORD guideline, and include the completed checklist as Supporting Information. Please add the following statement, or similar, to the Methods: "This study is reported as per the REporting of studies Conducted using Observational Routinely-collected Data (RECORD) guideline (S1 Checklist)." The guideline can be found here: https://www.record-statement.org/checklist.php

When completing the checklist, please use section and paragraph numbers, rather than page numbers which will likely no longer correspond to the appropriate sections after copy-editing.

RESULTS

Table 1: We suggest changing the table title to “Table 1 – Baseline study population statistics.”

Line 214 suggest: “Figure 1 […]” instead of “Fig 1[…]”.

Figure 2: Please add a close bracket in the figure legend changing “(Most deprive” to “(Most deprived)”. 

S2 Table/S4 Table/S6 Table: In your table description, you describe the results as adjusted by age (“Results from Cox proportional hazards models adjusted by age are reported as hazard ratios with 95% confidence intervals.”). However, the results presented are unadjusted by age as described in the main manuscript. Please revise accordingly. 

Lines 258-258 suggest, “However, amongst pre-school children, excluding those who may have attended primary school resulted in estimates of inequality that were no longer statistically significantly different across vaccination seasons.”

Lines 262-264: Please cite the supplementary tables S17 Table – S22 Table at the end of the sentence.

S21 Table/S22 Table: Please remove “Results also shown in Figure 2 in the main text.” from your table description.

Line 269: Supplementary table S17 appears to be wrongfully cited here. Please check and revise throughout your entire manuscript (S23 should be cited instead).

Figure 3: In Figure 3C, please show the axis beginning at zero. If this is not possible, please show a break in the axis.

Line 270-272/Line 284: Please add units to all numbers occurring in your manuscript when applicable (instead of ‘19.82’ it should be ‘19.82 percentage points’). Please check throughout your entire manuscript. 

Line 276/281/285/287: Supplementary tables S17 and/or S18 appear to be wrongfully cited here. Please check and revise throughout your entire manuscript (S23 and/or S24 should be cited instead).

DISCUSSION

Please remove the subheadings from your Discussion.

Lines 328-328 suggest: “Conversely, the lower risk of severe COVID-19 in children has led to considerable debate about the relative risks and benefits of vaccinating children and adolescents against COVID-19 [30,31], with the UK being slower than other high-income countries to offer vaccination to those under 18 years [32].”

Lines 330-331 suggest: “Our estimates of overall flu vaccine uptake within GM are consistently lower than national uptake estimates [6,33], likely in part due to above-average income deprivation within the region.”

Lines 344-346 suggest: “A strength of this study is the use of a dataset with almost population-level coverage for a large city region [24], along with robust measures of income deprivation that specifically capture the proportion of children or older adults living in poverty at a neighbourhood level [21].”

Lines 353-354 suggest: “Sensitivity analysis indicated that this is unlikely to have affected the results for primary school children or older adults.”

Lines 358-359 suggest: “While this could introduce bias, sensitivity analysis excluding those who died at any point in the study period indicated that this is unlikely to have substantially affected estimates of inequality.”

Line 367: Please change “over-represented” to “overrepresented”.

Line 383-384 suggest: “Another factor that may have affected inequalities in access to flu vaccination is the differential impact of long-term ill health due to long COVID [36] and pandemic-related delays in elective health care [37].”

REFERENCES

Please ensure that journal name abbreviations match those found in the National Center for Biotechnology Information (NCBI) databases (http://www.ncbi.nlm.nih.gov/nlmcatalog/journals), and are appropriately formatted and capitalised.

Where website addresses are cited, please specify the date of access.

SOCIAL MEDIA

To help us extend the reach of your research, please provide any Twitter handle(s) that would be appropriate to tag, including your own, your co-authors’, your institution, funder, or lab. Please detail any handles you wish to be included when we tweet this paper, in the manuscript submission form when you re-submit the manuscript.

Comments from Reviewers:

Reviewer #1: Alex McConnachie, Statistical Review

I thank the authors for their consideration of my original points, and I am happy with their responses, I have no further comments to make.

Reviewer #2: General comment

Thank you for inviting me to review this article. 

The revised draft of the paper reads very well. The authors have addressed in detail the previous set of reviews. 

Specific comments

L159 - The acronym SII is not defined. 

L176 - This sentence of the main analysis mentions 'excluding those who died before or during each season' while L180-181 says 'We then restricted the main analysis sample to those who remained alive throughout each vaccination season.', which is part of the sensitivity analysis. What is the difference between these two analyses?

Line 203-210: The numerical data reported do not exactly match those in Table 1. Please could you check them again. For example, 70426 children aged 2-3 years are reported while Table 1 indicates a total population of 70419; another example, 16613 children aged 2-3 years in most deprived group vs 16612 in the table. 

Line 264: Tables S17-22 are not cited. 

Line 267-87: Tables 23 & 24 should be cited instead of tables 17 & 18.

[LINK]

---

## [Editor Report · Decision Letter 3]

4 Aug 2023

Dear Dr. Watkinson,

Thank you very much for re-submitting your manuscript "Evaluating socioeconomic inequalities in flu vaccine uptake during the COVID-19 pandemic: A cohort study in Greater Manchester, England" (PMEDICINE-D-23-00344R3) for review by PLOS Medicine.

As we are planning to accept the paper for publication in the journal, there are still a number of outstanding minor changes that need to be addressed.

[LINK]

We look forward to receiving the revised manuscript by Aug 11 2023 11:59PM.   

Sincerely,

Alexandra Schaefer, PhD

Associate Editor 

PLOS Medicine

plosmedicine.org

Requests from Editors:

Please check your Competing Interest statement. In the Financial Disclosure statement, you have stated that one of the authors, Matt Sutton, is a Senior Investigator at the NIHR. As stated on our website, a “Relationship (paid or unpaid) with organizations and funding bodies including nongovernmental organizations, research institutions, or charities” is considered a non-financial competing interest. 

All authors must declare their relevant competing interests per the PLOS policy, which can be seen here:

https://journals.plos.org/plosmedicine/s/competing-interests

For authors with ties to industry, please indicate whether any of the interests has a financial stake in the results of the current study.

Please check your manuscript carefully for grammar, spelling and punctuation. Please consider proofreading by a native English speaker.

Title: Please revise your title to replace 'flu' with 'influenza'. We suggest “Evaluating socioeconomic inequalities in influenza vaccine uptake during the COVID-19 pandemic: A cohort study in Greater Manchester, England”.

l.54: In the last sentence of the Abstract Methods and Findings section, please describe the main limitation(s) of the study's methodology or change “A limitation” to “The main limitation”.

l.95: Please define ‘NHS’.

ll.123-126 suggest: “Greater Manchester is also a socioeconomically diverse area, including some of the least deprived neighbourhoods nationally, despite overall higher deprivation levels than the national average [21].”

l.204: Please introduce the abbreviation “CI” for confidence intervals at first use.

l.307: Please, for add square brackets open for “increased from 15.59 percentage points (95% CI 14.52,16.67]) in 2015/16 to 19.82 (95% CI 18.66,20.98]) percentage”. 

l.317: Please, add a square bracket open for “29.58,30.95])”. Please revise throughout your entire manuscript.

ll.443-444: Please provide a reference.

ll.477-478: Please remove the ‘Declaration of Interest’ statement. These details should only be provided in the according section of the online submission form.

Please check all figures and tables thoroughly.

Table 1: Define ‘GM’. The footnotes below the table do not correspond with the footnotes/asterisks in the table. Please revise and check throughout your entire manuscript.

Figure 1: Please add a unit for ‘uptake’ (%).

Figure 4/S24 Table: You describe the age-adjusted estimated difference in vaccine uptake between the least and most income-deprived areas as presented in percentage. However, the slope index of inequality is presented in percentage points. Please change the figure/table description and exchange “%” with “percentage points”.

Figure S2: Due to the size of the individual graphs, the visibility of the data presented in Figure S2 is poor. Please use a format in which the individual graphs are displayed larger.

S4 Table: Please remove “[…] and age 4 years for each season.” from the figure description as the results in this table are unadjusted for age (“The reference group is D10 (least deprived areas).”).

Table S1/S2/S7/S13/S17/S18: Please make sure to add an asterisk following IDACI in the tables.

S16 Table: A third reference group seems to be ‘people not clinically eligible for flu vaccination’. Please add in your table description.

[LINK]

---

## [Editor Report · Decision Letter 4]

1 Sep 2023

Dear Dr Watkinson, 

On behalf of my colleagues and the Academic Editor, Rebecca F. Grais, I am pleased to inform you that we have agreed to publish your manuscript "Evaluating socioeconomic inequalities in influenza vaccine uptake during the COVID-19 pandemic: A cohort study in Greater Manchester, England" (PMEDICINE-D-23-00344R4) in PLOS Medicine.

Prior to publication, we require that you make the following changes:

*Thank you for revising your Competing Interest/ Financial Disclosure statement. The paragraph starting with "MS is an NIHR Senior Investigator [...]" and ending with "[...] Member of five Study Steering Committees." only needs to be included in the Competing Interest section of the manuscript submission form. The statement “The funders had no role in study design, data collection and analysis, decision to publish, or preparation of the manuscript.” only needs to be included in the Financial Disclosure section of the manuscript submission form.

*Please define ‘NHS’ and 'NIHR' in the according manuscript submission form sections.

*Thank you for providing your RECORD checklist. Please replace the page numbers with paragraph numbers per section (e.g. "Methods, paragraph 1"), since the page numbers of the final published paper may be different from the page numbers in the current manuscript.

*Figure 1: Please add a unit for ‘age’ and ‘age group’ on the y-axis (years).

*Figure 2/3: Please define ‘D’ (decile).

*Figure 4: Please add a unit for ‘Slope index of inequality (SII)’ on the y-axis (percentage points).

*S Tables: In all supplementary tables, please ensure to define ‘D‘ along the acronym for the index.

*S1 Figure: Please define ‘NHS’.

*S2 Figure: Due to the space (first three graphs), please label each axis for the ‘older adults’ graphs on the right. In the figure description, please include the relevant indices (IDACI/IDAOPI) and define ‘D’ (decile).

*S16 Table: In the column header “Expanded age, plus age 17-49 who are clinically eligible”, please add ‘years’ (“Expanded age, plus age 17-49 years who are clinically eligible”).

*ll.189-190: Please additionally include the statement „All codes, algorithms, and code set validation used to define the populations, outcomes, exposures, and covariates can be found here: https://github.com/rw251/gm-idcr/tree/master/projects/025%20-%20Watkinson“ in the data availability section of the manuscript submission form.

*l.258: Please add ‘years’ following ‘across children aged four to eleven’.

We will carefully review whether these changes have been made prior to publication.

PRESS

Sincerely, 

Alexandra Schaefer, PhD

Associate Editor 

PLOS Medicine